# Clinical Impact of Sarcopenia and Inflammatory/Nutritional Markers in Patients with Unresectable Metastatic Urothelial Carcinoma Treated with Pembrolizumab

**DOI:** 10.3390/diagnostics10050310

**Published:** 2020-05-15

**Authors:** Takuto Shimizu, Makito Miyake, Shunta Hori, Kazuki Ichikawa, Chihiro Omori, Yusuke Iemura, Takuya Owari, Yoshitaka Itami, Yasushi Nakai, Satoshi Anai, Atsushi Tomioka, Nobumichi Tanaka, Kiyohide Fujimoto

**Affiliations:** 1Department of Urology, Nara Medical University, 840 Shijo-cho, Kashihara, Nara 634-8522, Japan; takutea19@gmail.com (T.S.); makitomiyake@yahoo.co.jp (M.M.); horimaus@gmail.com (S.H.); aburatani40@gmail.com (K.I.); chipanda2828@gmail.com (C.O.); housevillagenmu@yahoo.co.jp (Y.I.); tintherye@gmail.com (T.O.); y.itami.324@gmail.com (Y.I.); nakaiyasushi@naramed-u.ac.jp (Y.N.); sanai@naramed-u.ac.jp (S.A.); tomioka515@yahoo.co.jp (A.T.); sendo@naramed-u.ac.jp (N.T.); 2Department of Urology, Saiseikai Chuwa Hospital, 323 Ooazaabe, Sakurai, Nara 633-0054, Japan

**Keywords:** immune checkpoint inhibitor, Pembrolizumab, Sarcopenia, psoas muscle mass index, neutrophil-to-lymphocyte ratio, prognostic nutritional index, urothelial carcinoma, prognosis

## Abstract

Sarcopenia is a muscle loss syndrome known as a risk factor of various carcinomas. The impact of sarcopenia and sarcopenia-related inflammatory/nutritional markers in metastatic urothelial carcinoma (mUC) treated with pembrolizumab was unknown, so this retrospective study of 27 patients was performed. Psoas muscle mass index (PMI) was calculated by bilateral psoas major muscle area at the L3 with computed tomography. The cut-off PMI value for sarcopenia was defined as ≤6.36 cm^2^/m^2^ for men and ≤3.92 cm^2^/m^2^ for women. Neutrophil-to-lymphocyte ratio (NLR) ≥ 4.0 and sarcopenia correlated with significantly shorter progression-free survival (PFS) (hazard ratio (HR) 3.81, *p* = 0.020; and HR 2.99, *p* = 0.027, respectively). Multivariate analyses identified NLR ≥ 4.0 and sarcopenia as independent predictors for PFS (HR 2.89, *p* = 0.025; and HR 2.79, *p* = 0.030, respectively). Prognostic nutrition index < 45, NLR ≥ 4.0 and sarcopenia were correlated with significantly worse for overall survival (OS) (HR 3.44, *p* = 0.046; HR 4.26, *p* = 0.024; and HR 3.92, *p* = 0.012, respectively). Multivariate analyses identified sarcopenia as an independent predictor for OS (HR 4.00, *p* = 0.026). Furthermore, a decrease in PMI ≥ 5% in a month was an independent predictor of PFS and OS (HR 12.8, *p* = 0.008; and HR 6.21, *p* = 0.036, respectively). Evaluation of sarcopenia and inflammatory/nutritional markers may help in the management of mUC with pembrolizumab.

## 1. Introduction

Recently, immune checkpoint inhibitors (ICIs), including programmed cell death 1 (PD-1) inhibitors, such as nivolumab and pembrolizumab, have been shown to have promising effects in the treatment of various types of cancer [1,2,3,4,5]. Until the advent of ICI treatments, platinum-based cytotoxic chemotherapy was the standard treatment for unresectable metastatic urothelial carcinoma (mUC). Pembrolizumab has been widely used in Japan since 2017 for treating unresectable mUC that is resistant to conventional cytotoxic chemotherapy, including that using platinum. However, the response rate is still unsatisfactory. Only 15% to 25% of patients show sustained responses and benefits from treatment with ICIs [6,7,8,9]. Furthermore, patients sometimes experience serious adverse events (AEs), including immune-related adverse events (irAEs). Therefore, biomarkers for predicting the effects of ICIs have been sought out using various approaches. Such possible biomarkers reported in past studies have included programmed death ligand 1 (PD-L1), tumor mutation burden, and circulating immune markers [10,11]. However, these markers are incomplete at least when used alone [11]. Predictive and easily measurable biomarkers are needed to select patients for whom treatment with ICIs is suitable. 

Sarcopenia is a syndrome characterized by progressive and generalized muscle loss and weakness. It has been reported to be a predictor of perioperative complications and chemotherapy-related AEs, as well as a prognostic factor for various malignant tumors [12,13,14,15,16]. Tumor progression is triggered by an imbalance between the host’s antitumor immune response and the growth of the tumor. Therefore, the effectiveness of ICIs is expected to depend highly on the functionality of the host’s immune system. Body composition and the inflammatory and nutritional markers circulating in the blood are strongly related to the host’s immune status. Shiroyama et al. [17] reported that having sarcopenia status at baseline is a significant predictor of poorer outcomes in patients with advanced non-small cell lung cancer receiving PD-1 inhibitors. The neutrophil-to-lymphocyte ratio (NLR) and platelet-lymphocyte ratio (PLR) are easily measurable marker of inflammation and that have been reported as a prognostic factor in various cancer patients. An elevation of these markers reflects enhanced systemic inflammation and is associated with reduced tumor-specific immunity such as decreased tumor-infiltrating lymphocytes in the tumor [18,19]. Moller et al. [20] reported that blood immune cell biomarkers, such as the neutrophil-to-lymphocyte ratio (NLR), counts of human leukocyte antigen D-related (HLA-DR) low monocytes, and counts of dendritic cells (DC), are useful predictive markers for the outcomes of treatment of advanced non-small cell lung cancer (NSCLC) with ICIs. Furthermore, Ohba et al. [21] reported that patients with NSCLC treated with pembrolizumab having a high controlling nutritional status score (CONUT score) was significantly correlated with them having worse progression-free survival (PFS) or overall survival (OS). However, little is known about the clinical effects of skeletal muscle loss and blood immune cell biomarkers in patients with progressive mUC treated with PD-1.

We conducted a retrospective study to investigate the association of sarcopenia and inflammatory and nutritional biomarkers with treatment outcomes, including long-term responses to PD-1 inhibitors, in patients with advanced mUC who received previous chemotherapy.

## 2. Materials and Methods

### 2.1. Ethical Approval

This study was approved by the Institutional Review Board (IRB) of the Nara Medical University (Nara, Japan; Medical Ethics Committee ID: NMU-2153; approval date 18 March 2019) and complied with the 1964 Helsinki Declaration and its later amendments. As data for the study were obtained through a retrospective review, the need for informed consent was waived by the IRB. Personal information of the subjects and donors was anonymized when necessary, and the information was labeled with an identification code to make it possible to distinguish between individuals. De-identified patient data were then analyzed.

### 2.2. Patients Selection and Data Collection

Between December 2017 and August 2019, 29 patients who received pembrolizumab for chemotherapy-resistant advanced mUC in Nara medical university hospital were initially enrolled in this study. Two patients were excluded because of insufficient radiographic and laboratory examination, leaving the data from 27 patients to be included in our retrospective analysis. Clinicopathological variables and laboratory data at baseline and at all follow-up assessment time points were extracted from medical records for analyses of the time-course of changes after the initial administration of pembrolizumab. Figure 1A depicts the study design. 

### 2.3. Muscle Mass Index Values Calculated by Computed Tomography Scans

Unenhanced computed tomography (CT) images that were taken for diagnostic or follow-up purposes were inputted into the Volume Analyzer SYNAPSE VINCENT image analysis system (Fujifilm Medical, Tokyo, Japan) to calculate the body composition of patients in terms of the psoas muscle mass index (PMI). Measurements (areas) were normalized by patient height (m^2^) and expressed in units of cm^2^/m^2^ for between-subject comparisons. The following measures were obtained for analysis: area of the psoas major muscle at the caudal end of the third lumbar vertebra (L3) level (PMI, cm^2^/m^2^); and the total area of the abdominal skeletal muscle at the caudal end of the L3 level (cm^2^/m^2^). Representative images used for analyses of the abdominal muscle mass are shown in Figure 1B–D.

### 2.4. Definition of Sarcopenia

Hamaguchi et al. [22] reported diagnostic cut-off values for defining sarcopenia in Asians based on a study involving 541 adult living donors for liver transplantation. The cut-off value was defined as a PMI < 6.36 cm^2^/m^2^ for men and <3.92 cm^2^/m^2^ for women. Our study used this definition in all analyses, as did previous studies [17,23]. 

### 2.5. Measurement of Inflammation-and Nutritional Status-Based Makers

Baseline inflammation and nutritional status were evaluated using the following inflammatory markers: C-reactive protein (CRP), NLR, and PLR; and using the following nutritional status-based markers: geriatric nutritional risk index (GNRI), prognostic nutritional index (PNI), and controlling nutritional status score (CONUT). These markers’ values were calculated using laboratory data of patient’s charts obtained over a period of <30 days prior to the initial administration of pembrolizumab. The inflammatory markers were calculated using the ratio of each cell count per volume (cells/mm^3^). The GNRI was calculated using the following formula: 14.89 × serum albumin (g/dL) + 41.7 × current body weight (kg)/ideal body weight (kg) (ideal body weight was calculated as: height × height (m^2^) × 22) [24]. The PNI was calculated using the following formula: 10 × serum albumin (g/dL) + 0.005 × total lymphocyte count (cells/mm^3^) [23]. The CONUT score was determined on the basis of the serum albumin, peripheral lymphocyte count, and T-cholesterol levels, as described in Appendix A [25]. CRP, cholesterol, and albumin were measured by LABOSPECT006 (Hitachi High-Tech, Tokyo, Japan) and peripheral blood cells were measured by UniCel^®^DxH800 (BECKMAN COULTER, Brea, CA, USA). The cut-off value for each marker was determined using receiver operating characteristic (ROC) curve analysis, with all deaths defined as “events”. In addition to the sarcopenia status and inflammatory and nutritional status-based makers, we assessed age, Eastern Cooperative Oncology Group-performance status (ECOG-PS), body mass index (BMI), primary site of urothelial carcinoma, total courses of chemotherapy performed before pembrolizumab administration, and presence or absence of lung/liver metastasis as additional potential predictor variables. 

### 2.6. Follow-Up and Time-Course of Changes in Variables after Infusion of Pembrolizumab

Follow-up was performed after 1 month (1M) and then again after 3 months (3M). Oncological evaluations included physical examinations, routine blood tests, urine cytology, and CT scans of the chest, abdomen, and pelvis. Time-course changes from baseline in nutritional, inflammatory, and muscle mass-based markers were retrospectively observed, and were calculated by absolute value and relative value analysis, with the values of baseline data set as 100%. Response to treatment was assessed by radiological evaluation according to Response Evaluation Criteria in Solid Tumors (RECIST), version 1.1 [26]. Progression was defined as tumor enlargement based on the assessment of RECIST-based imaging, or when there was clinically obvious disease exacerbation. The period from the first dose of pembrolizumab to progression or death from any cause was calculated as the time to PFS or OS.

### 2.7. Statistical Analysis

Statistical analyses were performed and figures were plotted using GraphPad Prism 7.0 (GraphPad Software, San Diego, CA, USA). Fisher’s exact test or the Mann–Whitney U test was used for statistical analysis. The interrelationship between each marker was examined by calculating values of Spearman’s rank correlation coefficient between their values. To identify prognostic factors for PFS and OS, univariate and multivariate analyses were performed via logistic and Cox regression analyses, respectively, using IBM SPSS software, version 21 (SPSS Inc., Chicago, IL, USA). Two-sided tests were used in all cases, and a *p*-value < 0.05 was considered statistically significant in all analyses.

## 3. Results

### 3.1. Patient Characteristics and the Relationship with Sarcopenia

The baseline clinicopathological variable values, nutritional and inflammatory index values, and blood test results for the 27 patients included in our analysis are summarized in Table 1. The median number of doses was eight (range: 1–22). The median follow-up period after the administration of pembrolizumab in our survival analysis was 7 months (range: 1–20 months). Over the follow-up period, 15 patients (56%) died, and all of them died from mUC. Two patients (7.4%) showed a complete response (CR) to treatment, and eight patients (29.6%) showed a partial response (PR) to treatment, even though PRs were sometimes temporary. Thus, the overall response rate was 37.0%. The median time to response was 3.1 months (with 5.5 doses). Further, five patients maintained their disease status throughout the observation period. Five patients (18.5%) had a stable disease, and 11 patients (40.7%) had a progressive disease (PD) at the first or second radiological assessment point (1M or 3M). 

The median PMI (cm^2^/m^2^) before pembrolizumab administration was 5.80 and 3.81 in men and women, respectively. Fifteen patients (56%) in our study cohort were diagnosed with sarcopenia. Baseline variable values were compared between patients with and without sarcopenia (Table 1). Sarcopenia status was associated with a lower GNRI. 

### 3.2. Correlation Analyses of Biomarkers

We evaluated the correlation between the values of different inflammation-, nutritional status-, and muscle mass status-based biomarkers. Spearman’s correlation coefficient analysis was performed using selected continuous variables such as age, BMI, GNRI, PNI, CONUT score, NLR, PLR, skeletal muscle index (SMI), and PMI. Figure 2 summarizes the *p*-values and the Spearman’s ρ values obtained in these analyses. Among the nutritional status-based markers (GNRI, PNI, and CONUT score), PNI showed a significant correlation with both of the other markers of this type (*p* < 0.01). However, there was no correlation between the GNRI and CONUT scores. Among the inflammation-based markers, NLR and PLR were significantly correlated with each other (*p* < 0.01). Furthermore, PNI and CONUT score values were significantly correlated with these inflammatory markers (*p* < 0.001). Finally, the muscle mass status-based markers SMI and PMI significantly correlated with each other (*p* < 0.01). However, there was no significant correlation between the muscle mass status-based markers and any of the inflammatory and nutritional markers.

The following nine markers were compared: age, body mass index (BMI), geriatric nutritional risk index (GNRI), prognostic nutritional index (PNI), controlling nutritional status score (CONUT score), neutrophil–lymphocyte ratio (NLR), platelet-lymphocyte ratio (PLR), skeletal muscle index (SMI), and psoas muscle index (PMI).

### 3.3. Prognostic Values of Nutritional-, Inflammation- and Muscle Mass-Based Markers

Univariate and multivariate analyses were performed to determine the best prognostic factors at the first administration of pembrolizumab. Nutritional status-, inflammation-, and muscle mass-based markers were used to predict PFS and OS (Table 2 and Table 3, respectively). Kaplan–Meier curves for PFS and OS for the entire patients are shown in Figure 3A,B, respectively. Univariate analysis of PFS data revealed that an NLR ≥ 4.0 and sarcopenia status were statistically significant negative prognostic factors for PFS (*p* = 0.0020 and *p* = 0.027, respectively) (Figure 3C,E). Among the inflammation-based markers, only NLR was observed to be a statistically significant predictor of PFS, and CRP and PLR were not. In addition, none of the nutritional status-based markers were statistically significant predictors for PFS. Multivariate analysis of PFS data revealed that an NLR ≥ 4.0 and sarcopenia status were independent negative prognostic factors for PFS (*p* = 0.0025 and *p* = 0.030, respectively).

Univariate analysis of OS data revealed that an NLR ≥ 4.0 and sarcopenia status were statistically significant negative prognostic factors for OS (*p* = 0.0024 and *p* = 0.0012, respectively) (Figure 3D,F). Similar results were obtained for patients having an ECOG-PS ≥ 2, liver metastases, and a PNI < 45 (*p* = 0.0040, *p* = 0.0040, and *p* = 0.0046, respectively). Similar to PFS, inflammatory markers CRP and PLR were not observed to be statistically significant predictors of OS. Meanwhile, among the nutritional status-based markers, only PNI was observed to be a statistically significant prognostic factor for OS. Multivariate analysis of OS data revealed that sarcopenia status was an independent negative prognostic factor for OS (*p* = 0.0026).

PFS and OS were estimated using the Kaplan–Meier method. Overall, the median PFS of patients in our study was 4.0 months and the estimated PFS rate at 12 months was 15.8%, and the median OS was 7 months and the estimated OS rate at 12 months was 47.2% (Figure 3A,B). Patients with high NLR values and sarcopenia had significantly poorer PFS and OS than those with low NLR values and without sarcopenia (Figure 3C–F).

PFS, progression-free survival; OS, overall survival; NLR, neutrophil-to-lymphocyte ratio.

### 3.4. Time-Course of Change on Nutritional-, Inflammation- and Muscle Mass-Based Markers

The median (and range) points of maximal change after 1 month from baseline for PNI, CONUT, NLR, PLR, SMI, and PMI were −5.8% (−31.4 to +16.3%), 0 (−2 to +3 points), +17.1% (−58 to +215%),+12.8% (34 to +241%), −4.7% (35 to +31%), and −4.2% (−20 to +15%), respectively. The cut-off values for changes 1 month after the administration of pembrolizumab were determined based on median values as follows: −5% (PNI), +1 point (CONUT score), +15% (NLR), +15% (PLR), −5% (SMI), and −5% (PMI). 

Univariate analysis of PFS data revealed that a >15% increase in PLR and a ≥5% decrease in PMI were statistically significant negative prognostic factors for PFS (*p* = 0.013 and *p* = 0.0009, respectively) (Appendix A). Conversely, a ≥10% decrease in PNI, ≥1 increase in CONUT score, ≥15% increase in NLR, and ≥5% decrease in SMI were not associated with poorer survival outcomes (Appendix A). Multivariate analysis of PFS data revealed that a ≥5% decrease in PMI was an independent negative prognostic factor for PFS (*p* = 0.008) (Table 4).

Univariate analysis of OS data revealed that a ≥1-point increase in CONUT score, ≥15% increase in NLR, ≥15% increase in PLR, and ≥5% decrease in PMI were statistically significant negative prognostic factors for OS (*p* = 0.024, *p* = 0.020, *p* = 0.026, and *p* = 0.014, respectively) (Appendix A). However, a ≥10% decrease in PNI and a ≥5% decrease in SMI were not associated with poorer survival outcomes. Multivariate analysis of OS data revealed that a ≥5% decrease in PMI was an independent negative prognostic factor for OS (*p* = 0.036) (Table 5).

### 3.5. Adverse Event Rates and Effects of Sarcopenia

The incidences of AEs (any grade) and irAEs are shown in Appendix A. In our study, 12 (44%) patients experienced an AE, and 10 (37%) patients experienced an irAE. Five (19%) patients experienced high-grade (grade III–IV) AEs, all of which were irAEs. Univariate analysis and Fisher’s test were performed to investigate the association between sarcopenia and the incidence of irAEs (Appendix A). Sarcopenia status did not significantly correlate with irAE incidence. However, patients without sarcopenia had a greater tendency to develop high-grade irAEs, although this trend was non-significant (univariate analysis, *p* = 0.10).

## 4. Discussion

In this study, we investigated the impacts of sarcopenia and different inflammatory and nutritional markers on the efficacy of the use of pembrolizumab to treat unresectable metastatic urothelial cancer (mUC) in clinical practice. The evaluation of these markers before and during pembrolizumab administration may help in the management of mUC.

Initially, PD-L1 expression and tumor mutation burden were expected to be markers useful for predicting the effects of ICI, but their usefulness was actually limited and insufficient. Therefore, liver metastasis and NLR have recently attracted attention as markers that are easy to use in clinical practice [27,28]. In addition, skeletal muscle loss, which is often reported as a predictor of perioperative complications and chemotherapy-related AEs, and as a prognostic factor for various malignant tumors, has been reported to be associated with poorer OS and the occurrence of early acute limiting toxicity in melanoma and NSCLC treated with ICIs [17,29,30]. We therefore speculated that skeletal muscle loss could be a predictor of ICI treatment efficacy in mUC. We expected that skeletal muscle loss could be strongly reflected in the levels of inflammation and nutritional markers in the blood. However, in this study, skeletal muscle loss was not significantly correlated with any of the assessed inflammatory or nutritional markers. One reason for this may be that these markers are sensitive to local changes, whereas skeletal muscle loss is a consequence of more long-term effects. However, our group has previously reported the value of nutritional markers as prognostic factors for urothelial carcinoma, and as factors that lead to chronic kidney disease (CKD) after donor nephrectomy [13,14,15,23]. Furthermore, the relationship of inflammation, nutritional, and muscle status with PD-1 inhibitor treatment efficacy has not yet been evaluated well in patients with urothelial carcinoma. To the best of our knowledge, this is the first study to investigate these issues. In this study, the prevalence of sarcopenia was 56%, which was higher than the prevalence of sarcopenia in the preoperative baseline for urothelial carcinoma that our group previously examined [13,14,15]. This may be because of the decreased activities of daily living (ADL) and progression of cachexia associated with the progression of tumor growth and previous chemotherapy. In the pembrolizumab group in the KEYNOTE 045 trial [3], the percentage of patients with an ECOG-PS ≥ 2 was 0.7%, and the percentage of patients with two or more previous chemotherapy doses was about 20%. In contrast, our study had a higher percentage of patients with both an ECOG-PS ≥ 2 and ≥2 prior chemotherapies (Table 1). 

In the KEYNOTE 045 trial, the median PFS was 3.3 months and the estimated PFS rate at 12 months was 16.8% in the pembrolizumab group. Furthermore, the median OS in the pembrolizumab group in that trial was 10.3 months and the estimated OS rate at 12 months was 43.9% [3]. In our study, the median overall PFS of patients was 4.0 months and the estimated PFS rate at 12 months was 15.8%, and the median OS was 7 months and the estimated OS rate at 12 months was 47.2% (Figure 3A,B). These results are not very different from the results of the previously cited trial.

In contrast, we focused on levels of various biomarkers in the blood in the present study. Univariate analysis showed significant differences in both PFS and OS as a result of NLR values and sarcopenia status. Furthermore, sarcopenia remained an important prognostic factor for both OS and PFS in multivariate analyses. However, NLR only remained a prognostic factor in the multivariate analysis for PFS (Table 2 and Table 3). PFS was significantly shorter in patients with an NLR ≥ 4.0 than in those with an NLR < 4.0 (*p* = 0.020) (Figure 3C). Furthermore, OS was significantly shorter in patients with an NLR ≥ 4.0 than in those with an NLR < 4.0 (*p* = 0.024) (Figure 3D). This is consistent with the findings of a study by Moller et al. [20], who examined 35 patients with NSCLC treated with ICIs and reported that those with a high NLR at baseline had a poorer prognosis for both PFS and OS than those with a low NLR. They also reported that counts of HLA-DR^low^ monocytes and DCs at baseline are useful for predicting patients’ response to ICI therapy. Furthermore, they showed that a decrease in the NLR and/or HLA-DR^low^ monocyte count, and an increase in total DC frequencies during ICI therapies, was correlated with improved therapeutic responses and prolonged OS. In this study, we did not evaluate monocyte or DC counts, but did evaluate inflammatory biomarkers, including NLR and PLR, during pembrolizumab therapy. In addition, the rate of change in each biomarker over time and the prognosis for survival one month after the start of treatment were examined herein. Table 4 and Table 5 present the results of our univariate and multivariate analyses of the correlations between changes in various markers and PFS and OS at 1 month (1M) after the first pembrolizumab treatment. In these analyses, it was observed that increases in NLR and/or PLR were correlated with a poorer PFS and OS. Although increases in NLR were not statistically significant predictors of PFS, the same trend was observed in these data. Interestingly, changes in CRP after 1 month were not correlated with changes in the PFS or OS, but it was found that the change in CRP was more strongly correlated with those in the PFS and OS after 3 months (3M) (Appendix A). However, given the short observation period in this study, it is unlikely that it would be possible to predict the treatment effect based on these results, but rather the results likely represent the impacts of the disease itself. Therefore, we also assessed the loss of muscle mass and its impacts. PFS was significantly shorter in patients with sarcopenia than in those without sarcopenia (*p* = 0.027) (Figure 3E). Further, OS was significantly shorter in patients with sarcopenia than in patients without sarcopenia (*p* = 0.012) (Figure 3F). These results are in line with those reported by Shiroyama et al. [17] for NSCLC. They assessed 42 patients with NSCLC treated with ICIs, and reported that those without sarcopenia had a higher overall response rate (40.0 vs. 9.1%; *p* = 0.025) and 1-year PFS rate (38.1 vs. 10.1%) compared to those with sarcopenia [17]. In addition, we performed a longitudinal study of each studied biomarker, including body composition index values. We found that decreased PMI (i.e., an exacerbation of sarcopenia) was a poor prognostic marker for PFS and OS in both univariate and multivariate analyses (Table 4 and Table 5
Table 4; Table 5). Daly et al. [30] reported that sarcopenia and loss of muscle mass were significantly correlated with the occurrence of more severe AEs, and loss of muscle during four cycles of ipilimumab treatment is a risk factor for OS. As noted above, cross-sectional and longitudinal studies of body composition have been conducted in studies of other types of cancer treated with ICIs. However, there have been no such studies done on urothelial carcinoma as far as we know. In addition, we investigated the relationship between sarcopenia and the incidence of AEs. Univariate analysis and Fisher’s test were performed on these data (Appendix A), and we found that patients without sarcopenia tended to more often experience irAEs based on the univariate analysis results (*p* = 0.10). It has previously been reported that the treatment effect of ICIs is high when irAEs are observed [31]. Such irAEs are thought to represent the effects of activated T-cells. Activated T-cells show anticancer activity and contribute to autoimmune toxicity. Patients without sarcopenia may have higher immune activity, and are thus more likely to develop irAEs than those with sarcopenia.

Finally, we examined the levels of nutritional markers in the blood of patients. The results of our cross-sectional study revealed that OS was significantly shorter in patients with a PNI < 45 than in those with a PNI ≥ 45 (*p* = 0.046) (Table 3), and our longitudinal study revealed that an increased CONUT score value was a poor prognostic marker for OS (*p* = 0.024) (Table 4). However, these results were not retained in the multivariate analysis. Furthermore, GNRI, PNI, and CONUT scores were not found to be good prognostic factors for PFS. In a study of 32 patients with NSCLC treated with pembrolizumab, Ohba et al. [21] reported that high CONUT scores were significantly correlated with poorer PFS and OS. Shoji et al. [32] also assessed 102 patients with NSCLC who were treated with ICI therapy, and reported that PNI levels before treatment were significantly associated with the response of patients to ICI therapy. They found that PNI was an independent prognostic factor for both PFS and OS [32]. Nutritional status was expected to be correlated with sarcopenia status and inflammatory marker levels and to be strongly correlated with tumor-host immune responses. However, only some markers were correlated with one another in this study (Figure 2), and some markers were only correlated with OS in the univariate analysis of data from both cross-sectional and longitudinal examinations. This may have been largely due to the small size of the studied population.

In our study, we found that muscle, inflammatory, and nutritional markers have the potential to be used as biomarkers in patients with advanced mUC who were treated with pembrolizumab. 

These findings may be useful for improving the management of metastatic urothelial cancer, and should be considered as generating hypotheses to be tested in future studies. Furthermore, to the best of our knowledge, this is the first study to evaluate the use of sarcopenia status and inflammatory and nutritional markers as biomarkers in patients with advanced mUC who were treated with pembrolizumab.

The limitations of our study need to be acknowledged in the interpretation of our results. These include: the fact that the data used came from a single institution; the retrospective nature of the study, including the potential selection bias resulting therefrom; and the fact that analyses were done with a relatively small sample size and short follow-up duration. Furthermore, there is a lack of information regarding PD-L1 expression because PD-L1 status has not routinely been checked in Japan. A prospective study should be performed to overcome these limitations.

## 5. Conclusions

Our findings revealed that evaluation of sarcopenia and inflammatory and nutritional markers before and during pembrolizumab administration may help in the management of unresectable metastatic urothelial carcinoma. Nevertheless, further multicenter prospective research is required to validate these findings.

## Figures and Tables

**Figure 1 diagnostics-10-00310-f001:**
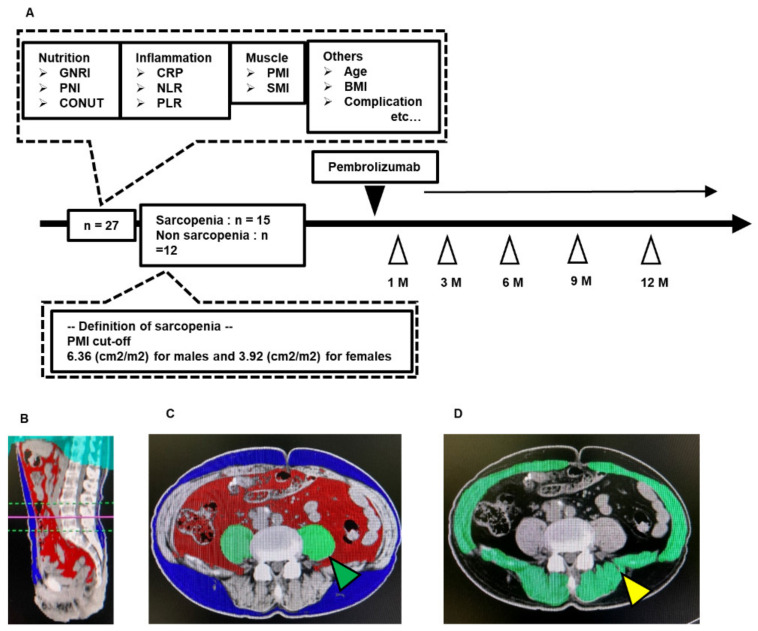
Schematic representation of study workflow (**A**) and representative images used for analyses of muscle mass-based markers (**B**–**D**). This study included 27 patients with mUC who received pembrolizumab at our institute between December 2017 and August 2019. Information regarding the pretreatment conditions was obtained retrospectively by reviewing the medical records, laboratory blood tests, and CT imaging. The PMI cut-off values of Hamaguchi et al. [20] were used to classify 15 patients as having sarcopenia and 12 patients as not having sarcopenia. Various biomarkers were evaluated before and during the administration of pembrolizumab. The Volume Analyzer SYNAPSE VINCENT image analysis system was used to reconstruct three-dimensional (3D) images as follows: sagittal plane of the psoas muscle area at the caudal end of the L3 level (**B**); coronal plane for assessment of the abdominal subcutaneous adipose tissue area (blue area), abdominal visceral adipose tissue area (red area), and psoas muscle area (green area indicated by green arrow) (**C**), and the skeletal muscle area (green area indicated by the yellow arrow) (**D**). UC, urothelial carcinoma; CT, computed tomography; GNRI, geriatric nutritional risk index; PNI, prognostic nutritional index; CONUT score, controlling nutritional status score; CRP, C-reactive protein; NLR, neutrophil-to-lymphocyte ratio; PLR, platelet-lymphocyte ratio; SMI, skeletal muscle mass index; PMI, psoas muscle mass index.

**Figure 2 diagnostics-10-00310-f002:**
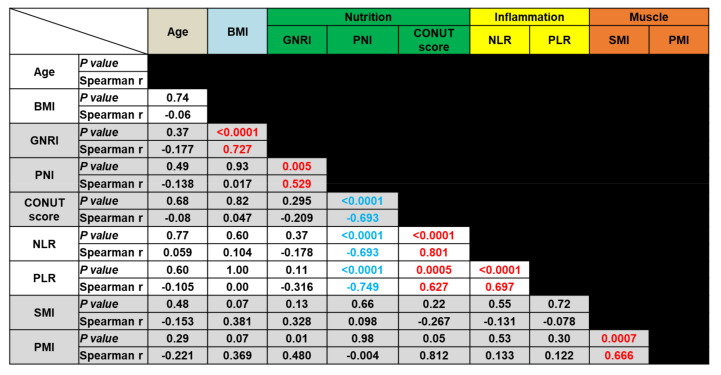
Correlations between nutritional status-, inflammation-, and muscle mass-based markers.

**Figure 3 diagnostics-10-00310-f003:**
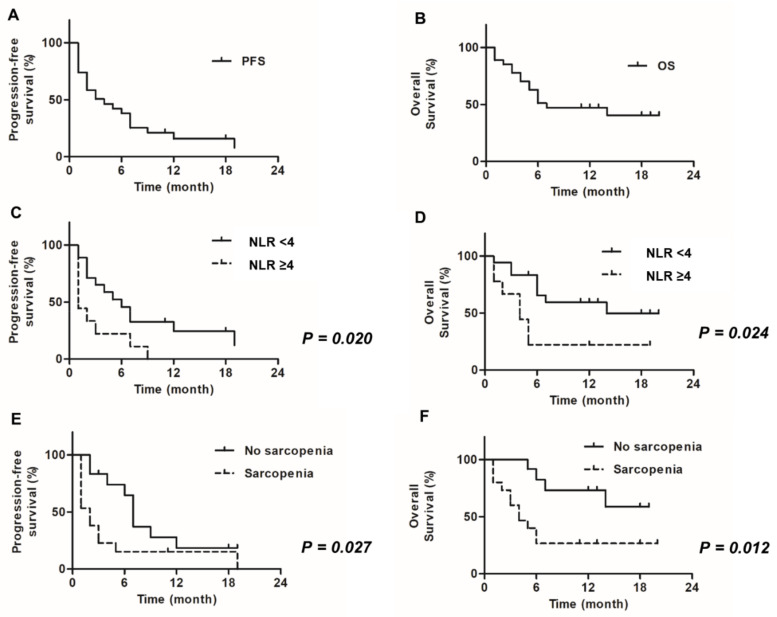
Progression-free survival (PFS) and overall survival (OS) probabilities. PFS and OS were estimated using the Kaplan–Meier method. Overall, the median PFS of patients in our study was 4.0 months and the estimated PFS rate at 12 months was 15.8%, and the median OS was 7 months and the estimated OS rate at 12 months was 47.2% (**A**,**B**). Patients with high NLR values and sarcopenia had significantly poorer PFS and OS than those with low NLR values and without sarcopenia (**C**–**F**). PFS, progression-free survival; OS, overall survival; NLR, neutrophil-to-lymphocyte ratio.

**Table 1 diagnostics-10-00310-t001:** Clinicopathological information on the first administration of pembrolizumab.

Variables		Total, *n* (%)	Non-Sarcopenia, *n* (%)	Sarcopenia, *n* (%)	*p* Value
Total		27 (100%)	12 (44%)	15 (56%)	
Age	Median (range)	73 (52–82)	69 (52–82)	74 (68–82)	0.1 ^†^
Sex	Male Female	23 (85%) 4 (15%)	11 (92%) 1 (8%)	12 (80%) 3 (20%)	0.61
ECOG-PS	0–1 ≥2	15 (56%) 12 (44%)	8 (67%) 4 (33%)	7 (47%) 8 (53%)	0.44
BMI (kg/m^2^)	Median (range)	21.9 (17.3–29.4)	23.1 (18.5–29.4)	20.6 (17.3–28.1)	0.07 ^†^
CCI	<8 ≥8	20 (74%) 7 (26%)	10 (83%) 2 (17%)	10 (67%) 5 (33%)	0.4
Autoimmune disease history	No Yes	26 (96%) 1 (4%)	12 (100%) 0 (0%)	14 (93%) 1 (7%)	1.0
Primary site	BT UTUC	15 (56%) 12 (44%)	7 (58%) 5 (42%)	8 (53%) 7 (47%)	1.0
Variant histology	No Yes	22 (81%) 5 (19%)	9 (75%) 3 (25%)	13 (87%) 2 (13%)	0.63
No. of prior chemotherapeutic regimens	1 ≥2	17 (63%) 10 (37%)	6 (50%) 6 (50%)	11 (73%) 4 (27%)	0.26
Evaluable lesion					
Primary site	No Recurrence Progression	19 (70%) 5 (19%) 3 (11%)	11 (92%) 0 (0%) 1 (8%)	8 (53%) 5 (33%) 2 (13%)	0.06
Regional lymph nodes	No Yes	11 (41%) 16 (59%)	4 (33%) 8 (67%)	7 (47%) 8 (53%)	0.7
Non-regional lymph node	No Yes	20 (74%) 7 (26%)	8 (67%) 4 (33%)	12 (80%) 3 (20%)	0.66
Lung	No Yes	12 (44%) 15 (56%)	7 (58%) 5 (42%)	5 (33%) 10 (67%)	0.26
Liver	No Yes	21 (78%) 6 (22%)	10 (83%) 2 (17%)	11 (73%) 4 (275)	0.66
Bone	No Yes	25 (93%) 2 (7%)	12 (100%) 0 (0%)	13 (87%) 2 (13%)	0.49
Nutrition markers					
GNRI	Median (range)	101.1 (72.9–122.8)	106.1 (98.9–121.1)	95.0 (72.9–122.8)	0.002 ^†^
PNI	Median (range)	48.5 (28.5–56.0)	48.5 (44–56)	46.0 (28.5–54.5)	0.09 ^†^
CONUT score	Median (range)	1 (0–8)	1 (0–6)	1 (0–8)	0.8 ^†^
Inflammation markers					
CRP (mg/dL)	Median (range)	0.54 (0.03–18.3)	0.25 (0.03–4.17)	0.66 (0.06–18.3)	0.15 ^†^
NLR	Median (range)	3.1 (0.9–37)	3.0 (0.9–7.0)	3.1 (1.2–37)	0.64 ^†^
PLR	Median (range)	185 (78–957)	178 (78–356)	214 (105–957)	0.48 ^†^
Muscle markers					
PMI at L3 (cm^2^/m^2^)	Median (range)	5.83 (2.21–8.94)	6.81 (5.99–8.94)	3.94 (2.21–6.27)	<0.0001 ^†^
Men	Median (range)	5.80 (2.21–8.94)	7.01 (6.37–8.94)	4.25 (2.21–6.27)	0.0002 ^†^
Women *	Median (range)	3.61 (2.95–5.99)	5.99	2.95, 3.61	NA
Follow-Up (month)	Median (range)	7 (1–20)	12 (5–19)	5 (1–20)	

ECOG-PS = Eastern Cooperative Oncology Group-Performance Status; BMI = body mass index; CCI = Charlson comorbidity index; BT = Bladder tumor; UTUC = Upper tract urothelial carcinoma; GNRI = geriatric nutritional risk index; PNI = prognostic nutritional index; CONUT = controlling nutritional status; CRP = C-reactive protein; NLR = neutrophil–lymphocyte ratio; PLR = platelet-lymphocyte ratio; PMI = psoas muscle index; L3 = at the level of the third lumbar vertebra; ^†^ Mann–Whitney U test. * Since only three female cases are included, the three cases presented are all cases.

**Table 2 diagnostics-10-00310-t002:** Univariate and multivariate analysis of background factors for PFS.

Variables		PFS					
		Univariate			Multivariate		
		HR	95% CI	*p* Value	HR	95% CI	*p* Value
Sex	Female Male	1 1.84	0.38–8.78	0.44			
Age	≥75 <75	1 0.59	0.23–1.56	0.29			
ECOG-PS	0 ≥2	1 2.55	0.95–6.86	0.06			
BMI	≥22 <22	1 1.43	0.56–3.61	0.45			
CCI	<8 ≥8	1 1.54	0.50–4.73	0.45			
Primary site	BT UTUC	1 1.75	0.68–4.49	0.24			
Variant histology	No Yes	1 0.78	0.26–2.36	0.66			
Number of platinumchemotherapy courses	≥4 ≤3	1 2.25	0.88–5.76	0.091			
Lung metastasis	No Yes	1 1.90	0.76–4.78	0.17			
Liver metastasis	No Yes	1 1.89	0.53–6.73	0.32			
GNRI	≥100 <100	1 1.95	0.76–4.95	0.16			
PNI	≥45 <45	1 2.10	0.75–5.93	0.16			
CONUT score	≤1 ≥2	1 1.60	0.59–3.92	0.38			
CRP	<0.5 ≥0.5	1 1.15	0.46–2.86	0.77			
NLR	<4.0 ≥4.0	1 3.81	1.23–11.7	0.020	1 2.89	1.10–7.03	0.025
PLR	<200 ≥200	1 1.94	0.74–5.08	0.18			
Sarcopenia	No Yes	1 2.99	1.14–7.85	0.027	1 2.79	1.14–7.32	0.030

HR = hazard ratio; CI = confidence interval; ECOG-PS = Eastern Cooperative Oncology Group-Performance Status; BMI = body mass index; CCI = Charlson comorbidity index; BT = Bladder tumor; UTUC = Upper tract urothelial carcinoma; GNRI = geriatric nutritional risk index; PNI = prognostic nutritional index; CONUT = controlling nutritional status; CRP = C-reactive protein; NLR = neutrophil–lymphocyte ratio; PLR = platelet-lymphocyte ratio.

**Table 3 diagnostics-10-00310-t003:** Univariate and multivariate analysis of background factors for overall survival (OS).

Variables		OS					
		Univariate			Multivariate		
		HR	95% CI	*p* Value	HR	95% CI	*p* Value
Sex	Female Male	1 1.84	0.38–8.78	0.59			
Age	≥75 <75	1 2.54	0.83–7.76	0.10			
ECOG-PS	0 ≥2	1 3.17	1.06–9.51	0.040	1 1.99	0.50–7.89	0.33
BMI	≥22 <22	1 1.73	0.60–4.97	0.031			
CCI	<8 ≥8	1 3.24	0.87–11.2	0.071			
Primary site	BT UTUC	1 1.61	0.54–4.80	0.39			
Variant histology	No Yes	1 0.80	0.23–2.86	0.81			
Number of platinumchemotherapy courses	≥4 ≤3	1 1.69	0.58–4.91	0.33			
Lung metastasis	No Yes	1 2.18	0.76–6.26	0.15			
Liver metastasis	No Yes	1 4.59	1.06–19.9	0.040	1 1.64	0.41–6.48	0.48
GNRI	≥100 <100	1 2.55	0.88–7.34	0.083			
PNI	≥45 <45	1 3.44	1.03–11.6	0.046	1 2.15	0.57–8.11	0.26
CONUT score	≤1 ≥2	1 1.58	0.54–4.58	0.4			
CRP	<0.5 ≥0.5	1 1.77	0.62–5.04	0.29			
NLR	<4.0 ≥4.0	1 4.26	1.21–15.0	0.024	1 1.22	0.21–7.20	0.82
PLR	<200 ≥200	1 2.37	0.83–6.78	0.11			
Sarcopenia	No Yes	1 3.92	1.34–11.5	0.012	1 4.00	1.18–13.6	0.026

HR = hazard ratio; CI = confidence interval; ECOG-PS = Eastern Cooperative Oncology Group-Performance Status; BMI = body mass index; CCI = Charlson comorbidity index; BT = Bladder tumor; UTUC = Upper tract urothelial carcinoma; GNRI = geriatric nutritional risk index; PNI = prognostic nutritional index; CONUT = controlling nutritional status; CRP = C-reactive protein; NLR = neutrophil–lymphocyte ratio; PLR = platelet-lymphocyte ratio.

**Table 4 diagnostics-10-00310-t004:** Univariate and multivariate analysis for correlation between PFS and changes of various markers 1M after the first Pembrolizumab.

Variable		PFS					
		Univariate			Multivariate		
		HR	95% CI	*p* Value	HR	95% CI	*p* Value
irAE	Yes No	1 0.98	0.26–3.75	0.98			
PNI decrease	<10% ≥10%	1 1.16	0.29–4.63	0.83			
CONUT score increase	No Yes	1 3.01	0.83–11.0	0.09			
CRP change	Low→Low High→Low the others	1 1.71	0.66–4.42	0.27			
NLR increase	<15% ≥15%	1 2.58	0.98–6.79	0.055			
PLR increase	<15% ≥ 15%	1 4.08	1.34–12.4	0.013	1 3.10	0.51–18.7	0.22
SMI decrease	<5% ≥ 5%	1 1.24	0.33–4.64	0.75			
PMI decrease	<5% ≥5%	1 17.1	3.18–91.7	0.001	1 12.8	1.91–85.4	0.008

HR = hazard ratio; CI = confidence interval; irAE = immune-related adverse events; PNI = prognostic nutritional index; CONUT = controlling nutritional status; CRP = C-reactive protein; NLR = neutrophil–lymphocyte ratio; PLR = platelet-lymphocyte ratio; SMI = skeletal muscle index; PMI = psoas muscle index.

**Table 5 diagnostics-10-00310-t005:** Univariate and multivariate analysis for correlation between OS and changes of various markers 1M after the first pembrolizumab.

Variable		OS					
		Univariate			Multivariate		
		HR	95% CI	*p* Value	HR	95% CI	*p* Value
irAE	Yes No	1 1.05	0.22–5.08	0.95			
PNI decrease	<10% ≥10%	1 2.18	0.52–9.04	0.28			
CONUT score increase	No Yes	1 8.77	2.16–35.6	0.024	1 2.88	0.39–21.5	0.30
CRP change	Low→Low High→Low the others	1 2.21	0.72–6.77	0.17			
NLR increase	<15% ≥15%	1 3.76	1.22–11.6	0.02	1 2.78	0.21–36.3	0.44
PLR increase	<15% ≥15%	1 3.74	1.17–11.9	0.026	1 1.96	0.20–19.4	0.56
SMI decrease	<5% ≥5%	1 3.15	0.75–13.3	0.12			
PMI decrease	<5% ≥5%	1 7.84	1.52–40.4	0.014	1 6.21	1.12–34.3	0.036

HR = hazard ratio; CI = confidence interval; irAE = immune-related adverse events; PNI = prognostic nutritional index; CONUT = controlling nutritional status; CRP = C-reactive protein; NLR = neutrophil–lymphocyte ratio; PLR = platelet-lymphocyte ratio; SMI = skeletal muscle index; PMI = psoas muscle index.

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
