# Peer review of "Clinical Impact of Sarcopenia and Inflammatory/Nutritional Markers in Patients with Unresectable Metastatic Urothelial Carcinoma Treated with Pembrolizumab"

_diagnostics, 2020, doi:10.3390/diagnostics10050310_

Round 1
Reviewer 1 Report
This is a well written and well organized study.
The below are the questions, suggestions, and concerns for the improvement of this manuscript;
- For the evaluation of sarcopenia, there are some markers are reported and confirmed except PMI, including inflammation markers, oxidative damage products, serum creatinine and urinary creatinine excretion, myostatin, atropin-1 (in muscle). Some are easy to use, and some are not. Is it possible to integrate additional markers for the sarcopenia evaluation for this study. It is more reliable to show the additional markers are also statistically related to the ICI response.
- Sarcopenia status is already known as a critical point for the prognosis of cancer patients (), and is this affecting to the ICI responsiveness or is this just a general prognosis marker for cancer patients ?
- There should be some introduction for NLR, including what this marker is indicating.
- For the responsiveness of ICI, mild inflammation is necessary and “cold immune” status shows ineffective response to ICI. On NLR, too high value is usually critically ill and too low is the “cold status”. Is there a possibility that the mild value of NLR (or other inflammatory scores) shows the highest response to ICI? If so, not cutting at score 4 of NLR, but cutting in low, mild and high grade of the score is more efficient scoring for the relationship with prognosis. If the authors are available to proceed, that might be valuable.
- Macrophage polarization (M1 vs M2) is also the another aspect indicating the immune status. If that is possible, please include the value or discuss at discussion part.
Reviewer 2 Report
I read the manuscript entitled “Clinical impact of sarcopenia and inflammatory/nutritional markers in patients with unresectable metastatic urothelial carcinoma treated with pembrolizumab” with the great interest.
The authors describe a study investigating the impact of prognostic nutrition index and sarcopenia on the overall survival in patients with unresectable metastatic urothelial carcinoma treated with pembrolizumab. The authors answer their primary outcome measures.
Title: The title is clear and accurate.
Abstract: The abstract is quite well written and definitions of the population are sufficiently well stated in the methods.
Introduction:
Could you add an information what is the percentage of unresectable metastatic urothelial carcinoma in patients presenting with invasive urothelial cancer in Japan?
Materials and methods: Methods are overall well described and I did not find any significant, flaws.
Discussion
I would like to encourage the authors to conduct prospective, more detailed study with assessment of PD-L1 expression.
Authors should also carefully revise the text, because it contains misspelled errors, for example:
Table S1. “Sever” should be change to “Severe”
Table S2. “Any immnune-related event” should be change to “Any immune-related event”
“Nausia” should be change to “Nausea”
“Hypothroidism” should be change to “Hypothyroidism”
“type 1 DM” should be more detailed describe as “Type 1 diabetes mellitus”
Reviewer 3 Report
The study is well conducted and has an impact on the current set of information.
The authors may need to provide or add the following to give the study a fullness.
1) In methodology, the authors need to provide info on the information on kist used for CRP, PLR and cholesterol, their source, and the part numbers.
The one question that the authors need to answer is apart from CRP what are the other inflammatory markers that were considered in the study? The authors may need to address how the changes in these inflammatory markers could effect pembrolizumab efficacy. Does it add resistance to the therapy?
The authors also need to address the caveats of this study apart from the sample size for future directions.
